# Caring for Family Caregivers of Geriatric Patients: Results of a Participatory Health Research Project on Actual State and Needs of Hospital-Based Care Professionals

**DOI:** 10.3390/ijerph17165901

**Published:** 2020-08-14

**Authors:** Theresia Krieger, Regina Specht, Babette Errens, Ulrike Hagen, Elisabeth Dorant

**Affiliations:** 1Department of Social Medicine, Faculty of Health Medicine and Life Sciences, Maastricht University, PO Box 616, 6200 MD Maastricht, The Netherlands; Theresia.krieger@maastrichtuniversity.nl; 2Helios Klinikum, An der Gontardslust 7, 57319 Bad Berleburg, Germany; regina.specht@t-online.de; 3Rhein-Maas Klinikum, Mauerfeldchen 25, 52146 Würselen, Germany; babe26.09@web.de (B.E.); ulrike.hagen@ct-west.de (U.H.)

**Keywords:** family caregiver support, geriatric patient care, actual state and needs assessment, health system development, participatory health research

## Abstract

Because of societal changes, family caregivers are becoming vital in long-term care provision for geriatric patients after discharge from hospital. Hospital-based geriatric care teams need tools to prepare and support family caregivers for their future caregiving role in the home environment. To explore the actual state and needs for implementing a suitable family caregiver support concept in a large geriatric hospital in Germany, a Participatory Health Research methodology was chosen. An academic investigator, assisted by a critical friend, facilitated all research steps. Geriatric care professionals joined as co-researchers and performed qualitative data collection using semi-structured interviews and focus group discussions. The entire co-research team took part in the thematic analyses. The existing family caregiver support was perceived as uncoordinated and incomplete, and a lack of knowledge about support programmes in the community was apparent. The needs regarding a comprehensive family caregiver support concept that acts on both individual caregiver as well as on system level, but also connects the two levels, were formulated. High grades of participation of hospital-based co-researchers could be achieved. A critical reflection on the research strategy revealed that the participatory methodology, although time-consuming, was perceived as a useful strategy within the hierarchically organized hospital.

## 1. Introduction

Family caregivers are considered as the backbone of the long-term care and support system [1,2]. Family or informal caregivers offer voluntary and unpaid physical, practical, and emotional care and/or support to a person with a disability in the home environment [3,4].

### 1.1. The German Situation

Recent figures from the he German federal statistical office (Destatis) show that, in Germany, 76% of all care to those with care needs is provided by family caregivers in their home environment [5]. Three to five Million German family caregivers provide unpaid care to 4–5 Million persons in need [6]. Most of these family caregivers are female, between 40–85 years old, and still in employment when younger than 65 years [7]. Family caregiving can be extremely stressful, especially in long-term care situations, e.g., when caring for a family member suffering from Alzheimer’s disease or a stroke [8,9]. Caregivers often experience a reduced quality of life, social isolation, loneliness, stress, fatigue, exhaustion, frustration, as well as a sense of helplessness [10,11,12]. A lack of choice of being a caregiver, lack of support, as well as financial strains and unfamiliar bureaucratic workload may also provoke caregiver burden [6,13]. Ideally, family members need to be prepared and educated before they take up their new role as informal caregiver [14,15]. In Germany, the official conditions for caregiver support are embedded in a law that supports the hospital discharge management (§39. Abs 1 S.9, SGB V) and the ‘Präventionsgesetz-PrävG’ [16]. Currently, different stakeholders provide different forms of caregiver support [6,7]. For example, health insurances provide online support, e.g., a “caregiving coach” [17], or group support, e.g., “caregiving courses” [18]. Communal services offer personalized “caregiving- and environmental counselling” (‘Pflege- und Wohnberatung’). However, these services are uncoordinated, unsatisfactorily utilized [19,20], or planned too late during the caregiving trajectory [21]. Especially in case of a sudden incident, such as a stroke, the organization of tailored family caregiver support should start already within the hospital or rehabilitation setting, i.e., before the patient is discharged. Acknowledgement of the prospective role of family members as part of the patient care trajectory together with well-timed, personalized support might also prevent the so-called ‘revolving door’ phenomenon [20,22].

### 1.2. Context of the Study

The Rhine-Maas-Clinicum is a large hospital located in Wuerselen, North-Rhine Westphalia, Germany, with a strong hierarchical organizational structure. The Rhine-Maas-Clinicum has a large geriatric department, with 64 acute geriatric beds and 28 rehabilitation beds spread over three wards. In 2016, 1550 patients, on average 82.7 years old, were admitted. Treatment, care, and medical support is provided by around 70 professionals, with a strictly patient-centered approach.

Staff members at the Rhine-Maas-Clinicum treat family caregiver issues in an unstructured, haphazard manner. The needs of family members, as prospective caregivers at home, are insufficiently acknowledged, although care professionals recognize the value of family members in the care trajectory of their patients.

Family caregivers have been shown to play an unnamed but critical role in the geriatric health care system, forming a so-called ‘shadow workforce’ [23]. They are an important source of information for geriatric care professionals, being the ones able to supply key pieces of medical information about their elderly care receiver in a timely manner [23]. However, they often experience strain, especially co-resident caregivers, and run the risk of getting depressed [24]. In a large meta-analysis, it was shown that receiving support, formal or informal, was related to better caregiver health [25]. Supporting them will secure the care to geriatric patients in the home environment after hospital discharge [26].

Recent recognition of the importance of the family as prospective caregivers for geriatric patients prompted the Rhine-Maas-Clinicum to join an academic research team from Maastricht University, The Netherlands, in setting up a Participatory Health Research (PHR) project to prepare the implementation of a new support programme for family caregivers, called Vade Mecum [27,28]. Vade Mecum aims to sustain health and wellbeing and prevent social exclusion and health inequality of geriatric patient-caregiver dyads. At the core of the concept is a new job profile: a hospital-based care employee with a special assignment to guide, educate and support family members of geriatric patients in becoming an informal caregiver [27]. A first step in designing a better system of geriatric care in the Rhine-Maas-Clinicum is to identify the status-quo and needs of geriatric care professionals to support them.

### 1.3. Aim of the Study

Before full implementation of the new family caregiver support concept in the Rhine-Maas-Clinicum, a pre-implementation step is necessary to gain insight in the status-quo activities and needs of the multidisciplinary team (MDT) regarding family caregiver support. Because of the haphazard way of engaging with family caregivers of geriatric patients by individual care professionals, little is known about how and what is needed to best deliver that support in a more organized way and to inform the hospital management on the best way of implementing the new holistic support concept.

In this article, we report on both the process and the results of a PHR study on actual state and needs of the MDT. At the core of the PHR approach is the delegation of decision-making power regarding the research process from academic researchers to a co-research team consisting of members from the MDT. This co-research team will be formed at the start of the study and is responsible for choosing the research design, participant profile, data collection methods, data analysis and reporting. The actual research question will be articulated by this co-research team during the PHR process.

## 2. Materials and Methods

### 2.1. Research Strategy

For this pre-implementation step, a participatory research strategy is a preferred choice. PHR has a longstanding tradition in public health research [29], and is viewed with increasing interest by researchers, decision-makers, and practitioners worldwide as a strategy to overcome the gap between academic researchers and professionals working in practice [30]. One of the principles of PHR is a power shift in the conduct of research from academic or ‘expert’ researchers to those stakeholders in the field that possess insightful knowledge and experience [31,32]. In addition to the generation of new knowledge, this approach also aims to empower those involved in the research process [33,34]. Co-researchers learn methods of scientific knowledge production and use the process to understand their own reality, whereas academic researchers understand the real needs of a practiced-based phenomenon [35]. Engaging members of the MDT as co-creator of new knowledge creates ownership of the outcomes [36].

### 2.2. Formation of the Research Team, Roles, Research Activities

Three categories of researchers were involved in this study: academic researchers (AR), a critical friend (CF) and co-researchers. The AR (first author) was responsible for: (1) facilitating and stimulating the PHR research process as “*external impulse provider*”, (2) contributing with scientific, technical and managerial knowledge, (3) drafting the research plan and monitoring the progress, and (4) guaranteeing a constant communication flow. The AR started the study by contacting all members of the MDT and explaining the participatory strategy.

During the research process, a ‘critical friend’ (CF, second author) supported the AR. A CF is a trusted professional who is able to ask proactive questions and give collegial feedback [37]. The CF in our study was serving as ‘listening ear’, stimulated reflection and continuous professional development of the AR [38,39].

In our study, members of the MDT were participating as co-researcher with researchers from outside the setting. Members of the MDT are the ones with experiential knowledge in the setting, have ideas about relevant research questions, the acceptability and appropriateness of the research methods and the interpretation of the findings. In principle, all members from all professional groups of the MDT working at the geriatric department of the Rhine-Maas-Clinicum, were eligible as member of the co-research team provided that they have contact with family caregivers. Research experience was not necessary. The head of the department, a medical doctor, served as gatekeeper.

The co-research team together with the AR and CF set the research agenda, formulate the research question, decide on which qualitative research methods to be used to collect the data, formulate selection criteria and recruit key informants in their peer group, collect the data, perform the data analyses, report the outcomes, and carry out all other necessary research activities to answer their research question [40,41]. The proceeding of this process is reported in the Results section Part 1: PHR process, and in Table 1.

### 2.3. Documenting and Reflection

A research management plan was developed to keep track of all research aspects and choices, and document who did what on what level of participation (Table 1). The AR outlined six phases beforehand: ‘orientation’, ‘setting-up’, ‘planning’, ‘data collection’, ‘analysis’, and ‘reporting’, according to the key principles developed for utilizing participatory methodologies in the co-creation and evaluation of public health interventions [42] The AR took lead in the preparation, organization, planning and communication of the project. As the iterative research process progresses and the roles change, contributions, activities and expected or planned outcomes will be inserted in the management plan to document the emerging study. In each phase, the participation grades of the co-researchers will be determined, and graded by using Cornwall’s participation typology, which ranges from 1 ‘co-option’, 2 ‘compliance’, 3 ‘consultation’, 4 ‘cooperation’, 5 ‘co-learning’, to 6 ‘collective action’ [43].

To evaluate the experiences with the PHR approach and critically reflect on the benefits and limitations of this research methodology within the setting, a brainstorming session with mind mapping will be carried out in the final meeting, when the co-researchers are still together as a group.

Regular meetings between the AR and the supervisor (last author) were organized to discuss important research issues and the progress of the subsequent steps, but also ways of ensuring rigor of the study [44,45]. The CF assisted the AR on keeping a reflective mindset, while the AR kept record of reflective talks with the CF. All research choices, activities and reporting in this manuscript were guided by the Standards for Reporting Qualitative Research [46].

### 2.4. Ethical Considerations

This participatory project aimed to assess the status-quo and needs of hospital personnel regarding support of family caregivers of geriatric patients, in preparation of their new to develop working practice. Participation as co-researcher or participant in one of the in-depth study parts was on a voluntary basis, no incentives were offered. Verbal informed consent was obtained from all individual study participants in the professional peer groups before data collection. The head of the department enabled participation during working hours and facilitated the research activities (e.g., providing materials and interview rooms). Ethical approval was given by the hospital management, approval from an ethics committee was not necessary.

Mutual respect, equality and inclusion, democratic participation, active learning, making a difference, and personal integrity were the ethical principles accompanying the entire PHR project [47]. Additionally, the following ‘ethics in practice’ principles were respected: (1) beneficence, (2) respect for autonomy, (3) justice, and (4) non-maleficence [48].

**Table 1 ijerph-17-05901-t001:** The project management plan of the PHR study on actual state and needs of the multidisciplinary geriatric team with respect to family caregiver support, ordered by phase in the life cycle. The activities and methodology and planned/expected outcomes were entered during the project.

Phase	Timing (month)	Participation and Contribution	Activity and Methodology	Planned/Expected Outcome
Orientation	1	AR takes the lead, MDT provides impulses	Start-up: Literature exploration Explorative talks 2-day field visit (interviews, observations)	Gaining a theoretical and practice-based understanding Starting engagement with decision makers and MDT Showing and raising interest within the department Starting topic list to be reflected on in phase 2
Setting-up	2–3	AR takes lead, MDT participates	Impulse presentation: (focus on family caregiving and caregiver needs)	Giving feedback concerning AR findings during orientation Raising awareness for caregiver support needs Stimulating the MDT to participate in the PHR process
		AR prepares basic research requirements MDT takes the lead	Kick-off workshop: Impulse lecture (Focus on PHR strategy) Interactive focus group discussion (brain storming, mind mapping)	Understanding the goals and ethical criteria of PHR ^§^ Exploring problem, expectations, and goals of the PHR project Formulating research question AR and members of the MDT who decide to participate as co-researchers are forming the research team
Planning	4–6	AR prepares list of potential methods for qualitative data collection CF stimulates reflection CR ^¶^ decide on data collection methodology	Project workshop 1: Impulse lecture (emphasis on advantages and disadvantages) Interactive focus group discussion	Discussing pros and cons of potential methods of qualitative data collection Sharing experiences with previous research and exploring practicability within the department and per peer group Stimulating continuous professional development Specifying participant subgroups for data collection Selecting appropriate qualitative data collection methodology, and assigning a method for each participant subgroup
Data collection	7–9	CR take the lead, AR and CF support the decision-making process	Project workshop 2: Explorative focus group discussion	Determining plan of action for qualitative data collection (Gantt chart) Designing a ‘question catalogue’ (53 questions regarding caregiver needs, information, knowledge, skills, resources, mandate, and support offer). Preparing a checklist for data collection
		CR take the lead, AF and CF support the process	Field work: within peer groups	Collecting qualitative data in each sub-group Offering technical, practical, and communicational support Supporting data storage
Analysis	10–13	AR leads the session CF provides practical impulses CR decide on further steps and new team composition	Project workshop 3: Interactive impulse session (emphasis on data analysis)	Gaining an understanding of the process of data analysis, required resources and team commitment Building a data analyzing team (from each profession one person is part of this team plus AR/CF)
		AF, CF and CR analyze data	Project workshop 4–5: Structured analyzing process	Analyzing qualitative data per subgroup Merging data from subgroups Formulating conceptual recommendations
Critical reflection on PHR	14	CR, CF and AF share their experiences	Final workshop (closing down): Brain storming followed by mind mapping	Critically discussing participation grades and perceived benefits and limitations of the PHR approach in general and within their own roles (CR, CF and AR)
Reporting	14	CR report on the findings AR and CF support the reporting process	Final workshop (closing down): Data presentation	Final report: actual state and needs of the MDT with respect to holistic family caregiver support
	14-ongoing	Individual members of research team	Individual reporting activities	Presenting the findings within the hospital Disseminating new knowledge on findings and PHR process (publication, conference, regional geriatric working group)

* Some fragments of the project up to the phase of data collection were published earlier [27]; ^¶^: CR = co-researchers; ^§^: From [47].

## 3. Results

First, in part 1, the PHR process will be presented per phase, accompanied by the management plan (Table 1). Next, both outcomes (actual state and needs) will be presented (part 2), and finally reflections on the research process will be addressed (part 3). The project started in February 2017 and ended in March 2018.

### 3.1. Part 1: PHR Process

#### 3.1.1. Orientation

The PHR life cycle started with the literature exploration, explorative talks with professionals (head of department, nurses, and social workers), and a 2-day field visit by the AR. A list with topics (current and previous caregiver support offers within the department and the hospital, actual engagement and potential interested professionals or stakeholders, resources and competences) was constructed to guide the first meeting with the prospective team members within the geriatric department.

#### 3.1.2. Setting-Up

The second phase started with an impulse lecture to raise interest to conduct a PHR study and, by doing this, to improve the caregiver support within the geriatric department. Members of the various professions in the geriatric department were invited as co-researchers. Careful attention was paid to ascertain representation of all levels of the hierarchical system within the hospital. Two medical doctors, two case managers, two social workers, two therapists (physiotherapy, speech therapist), and four nurses took part as ‘internal’ stakeholders. The AR with the CF completed the research team.

In the first workshop, the team was asked to reflect on their daily challenges in the interaction with family caregivers. System-related problems (knowledge gaps, lack of resources, coordination deficits), as well as caregiver-related problems (non-compliance, overburdening, communication deficits) were brought up. Additionally, communal support offers were perceived as fragmented, uncoordinated, and lacking a holistic approach. Caregivers were occasionally involved as partners in the hospital care process, but mainly for purposes of patient care. Caregiver burdens or needs were not assessed systematically at any time during the patient’s hospital stay.

Co-researchers also reflected on their own expectations and goals of the new support concept. They expected to allocate more time to support caregivers ‘tuned to caregiver’s individual needs’, scheduled in the normal working hours, to benefit from task simplification, to expand their own professional competence and get appreciation. Objectives were that family caregiver-patient dyads were given longer consultation times, as well as individualized and improved support within the hospital.

At last, guided by triggering questions from the AR, the main research question was formulated by the team: How does the multidisciplinary research team of the Rhine-Maas Clinic currently supports family caregivers, and what is needed for offering comprehensive support?

#### 3.1.3. Planning

In the third phase, the co-researchers first decided that a qualitative data collection method is appropriate to answer the research question. Next, they decided, based on the organizational structure within the department, on the groups of participants for the study. Six participant groups were indicated: medical doctors, nurses, physio-/occupational therapists, speech therapists, case management, and social services. The co-researchers chose data collection methods after an impulse lecture from the AR, who preselected and presented five possible methods: interviews, focus groups, storytelling, structured interview matrix and community mapping. For each method, advantages and disadvantages were discussed first, taking the existing experience within the geriatric setting into account. Interviews and focus groups seemed to be the best option.

Next, co-researchers were equipped to conduct the data collection within their own peer groups, with an instruction memo and a question catalogue prepared beforehand to guide and harmonize data collection within the different professions [27]. The AR monitored the data collection process.

#### 3.1.4. Data Collection

Study participants for the interviews and focus groups were recruited by co-research team members among their peers. In total, 31 individuals took part in the participatory data collection phase. Structured focus group discussions were organized by the medical doctors among the medical doctors (*n* = 8), nurses discussed with nurses (*n* = 10) and physio-/occupational therapists with their own peers (*n* = 5). Semi-structured interviews were held by the speech therapists (*n* = 3), case managers (*n* = 3), and social workers (*n* = 2). Each group organized their own meetings. Field notes were produced per group, and then distributed within the entire research team (co-researchers, CF, AR) to be analyzed.

#### 3.1.5. Analyses

A ‘data-analyzing team’ was formed, consisting of two medical doctors, two nurses, one occupational therapist, one physiotherapist and one social worker. The analyzing process was facilitated by the AR and CF. The ‘data-analyzing-team’ met three times for approximately 70 min per meeting and used a stepwise thematic analysis strategy [49,50] to analyze both the three focus group datasets as well as the three interview datasets. All members of the ‘data-analyzing team’ were familiarized with all six sets of field notes. Themes and subthemes were agreed upon per dataset first, after which the outcomes from the six data sets were merged, resulting in one final comprehensive multiperspective understanding of both the actual state and the needs for offering caregiver support within their setting.

### 3.2. Part 2: Outcomes

#### 3.2.1. Outcome 1: The Actual State of Caregiver Support

From the thematic analysis, three themes emerged: (1) professionals’ perception of the family caregivers, (2) actual state of professional caregiver support activities, and (3) weak spots or deficits in current caregiver support.

Ad (1): Professionals perceive family caregivers as an important interlinking person between the patient and the multidisciplinary team, but also as at risk of becoming overburdened by infrastructural issues, e.g., by changes in contact persons, or not being informed about rehabilitation and support offers. The social services said that: *“the role of caregivers is underestimated in our department”*. The physio- and occupational therapists observed that *“only ‘demanding’ caregivers are supported and ‘quiet’ relatives often fall through the grid”*. During their daily work, some professionals experience caregivers in general as *“under-supported”* and/or *“insufficiently informed”* [nurses; physio-/occupational therapy], while others even experience them as an *“encumbrance”* or *“added task in their daily work”* [case managers, medical doctors].

Ad (2): The intensity of contact, extent and moment of involvement varies greatly between the different professions. The existing caregiver support system is perceived as deficient, unstructured and uncoordinated: *“Currently, caregiver support is very chaotic”* [physio-/occupational therapy]; *“Caregiver counselling is rather inadequate, especially on medical and nursing topics, since there are no fixed office hours nor are individual appointments offered”* [social services]. However, all professionals provide some support on an individual basis: *“One medical doctor offers caregiver consultation talks from 3:00 pm to 4:00 pm”* [medical doctor]. Interdisciplinary consultations can be scheduled on request by the caregiver. The knowledge concerning caregiver-specific offers varies greatly among the professionals, and there is hardly any knowledge regarding the quality or acceptance of these offers. Within the hospital, two offers are known: (a) individual dementia caregiver counselling, which is provided by the clinic pastor; and (b) a peer group for dementia patients’ caregivers. Some professionals recommend external support offers to caregivers, e.g., health insurances, case managers or practical nursing skill courses (‘Pflegekurse’) provided by the communal services.

Ad (3): For the hospital management, the importance of family caregivers in the patient care trajectory is not recognizable. Caregiver support and counselling activities are not financed. Additionally, the number of available staff is so low that comprehensive caregiver support is difficult to organize. Caregiver support is insufficiently outlined in hospital job descriptions. Professionals feel frustrated and undervalued: *“There is little appreciation from the side of the management about our work with the relatives”* [medical doctor]. No caregiver support concept or guideline is implemented in the department or elsewhere in the hospital. Professionals lack important resources, such as time, staff, infrastructure (e.g., a counselling room, information materials), and internal support services. *“We have no reserved time slots for caregiver instruction”* [physio-/occupational therapy]; *“Our office is occupied by three people plus serves as the meeting room for caregiver consultation”* [social services]; *“Demonstration or training materials on site are either outdated or non-existent.”* [physio-/occupational therapy]).

Professionals recognize caveats in their caregiver-specific knowledge, skills and competences, blaming their vocational or academic education: *“We received no orientation on how to conduct a conversation with caregivers in our vocational education. We had to learn this in the job“* [medical doctor]; *“We possess some experiences, however our knowledge is still expandable”* [case management]. Professionals felt that insufficient attention is given to capacity building of staff concerning caregiver issues within the hospital. Although they can participate in external trainings, the conditions (work exemption, funding) vary greatly: *“You have to acquire the knowledge yourself”* [physio-/occupational therapy].

#### 3.2.2. Outcome 2: Needs of the Geriatric Team

After careful discussion, four themes emerged from the analyses: (1) the need for an inclusive caregiver support concept, (2) conceptual building blocks, (3) required resources, and (4) enabling conditions.

Ad (1): A holistic, needs-oriented (flexible), personal and professional caregiver support concept is required. *“We must find a system that involves relatives in a more structured way”* [social services]. The values of caregiver counselling must be taught to the entire multidisciplinary team. Counselling should involve: (a) an orientation on caregiver support offers within the department and beyond, e.g., communal support, (b) personalized information transfer, e.g., financial, medical and rehabilitative issues, and (c) practical support, e.g., instructions for care and mobilization: *“A mixture of good information about processes, illnesses and supply options* via *internet, flyers, group offers and individual advice is needed.”* [social services]; *“Relatives should know the duties of each professional group”* [physio-/occupational therapy]. Moreover, psychosocial support should be family-centered and better counselling facilities are needed. Maybe a self-help peer group, caregiver information evenings and group counselling sessions would be supportive: *“Some caregivers may be willing to meet other caregivers”* [nurses]; *“Group counselling is in some occasions helpful”* [social services].

Ad (2): To offer to caregivers relevant and valuable information, a number of options were mentioned: (a) a caregiver *“welcome folder”* containing general information (procedures, professions, contact persons); (b) profession-specific flyers presenting the different professions, processes and contact persons; (c) information boards within the department ‘advertising’ caregiver support; and (d) a ‘caregiver link’ on the hospital’s homepage leading to general and specific information and to support offers within the region. Caregiver support needs to be tailored and long-term: *“They need most support in the first days during the acute phase and a sound orientation on what to do when the patient will be discharged“* [nurses]. Creating a new job in the geriatric department, providing professional caregiver counselling, might be a solution. *“A permanent contact person for caregivers is needed”* [medical doctors]. This position requires professional experience, strong communication skills, system knowledge as well as special personality traits, e.g., empathy, openness: *“The ability to communicate and resolve conflicts, a degree of diplomacy, the ability to keep calm and pay attention to facts as well as active listening is crucial“* [social services]. Furthermore, *“outreach counselling is needed”* [social services]. A family caregiver assessment tool, self-administered by the caregivers before the first counselling, was proposed: *“A separate sheet to be completed by caregivers would be useful in many cases”* [social services]. The creation of an interprofessional network, which connects all stakeholders providing internal and external support, was suggested: *“Networking and participation in external events and organizations, e.g., Dementia Network Aachen, Alzheimer’s Society, other geriatric institutions is needed*” [social services]; *“We must think and act as a team”* [physio-/occupational therapy]; *“We need to create access to supportive and further-serving institutions“* [social services].

Ad (3): the need for resources were expressed: time, infrastructure (e.g., a counselling room within the department with a PC), equipment for family education, and more staff. *“The social services should be on the spot all day every day, as it has a large role in attending caregivers”* [medical doctor]; *“Fixed counselling hours should be offered, e.g., caregiver friendly time slots, and the contact persons’ name and accessibly should be provided”* [social services]; *“Adequate demonstration and training tools are required for caregiver and patient training”* [physio-/occupational therapy]; *“Supervision and case review sessions should be organized”* [social services]. Needs-oriented internal or external capacity building activities were advised: *“Caregiver support”* [social services]; *“Communication strategies”* [case management]; *“De-escalation training or how to best deal with overburdened relatives”* [physio-/occupational therapy]; *“The nursing school curriculum should extend to cover caregiver issues”* [medical doctors]. Capacity building should be financially compensated.

Ad (4): the need for enabling conditions within the hospital (e.g., political will, focus setting) was expressed. “Due to the new discharge management law, we have a take care now also for the patient beyond the inpatient stay. Here, more activities are needed across all professional groups within the department, e.g., social services, psychologists” [medical doctors]. To do this, an increased appreciation and attention for caregiver support in the geriatric department by the hospital management is required. “We need increased attention to the special concerns of the geriatric department” [social services]. The management must encourage a caregiver-friendly climate within the hospital and should trigger the development of a comprehensive caregiver concept. “There is a gap in the system that needs to be fixed as soon as possible” [medical doctors].

### 3.3. Part 3: Critical Reflection

In a separate part of the final meeting, the entire research team reflected on their experiences with the PHR approach.

#### 3.3.1. The PHR Approach

Overall, the co-researchers, CF and AR experienced this PHR approach as ‘inspiring and motivating’. The AR experienced the sharing of power as mutual learning process that enabled a true engagement of the co-researchers. However, compared to classic top-down research, conducting PHR for the first time in a hierarchic setting requires a great deal of time and investment in communication and project management. The CF recognized the changing team dynamics and was positively surprised how her comments were stimulating and motivating the team during the different phases. Beside the positive effects, she noticed that it was quite time consuming for the entire team. Initially, co-researchers perceived this project as “black box”, especially regarding its outcome and practicability within this hospital. However, the preconditions for the project were set as the head of the department was in favor of the PHR approach. Finally, co-researchers concluded that *“participatory research works. It magnifies the knowledge and has changed us”* [medical doctors].

#### 3.3.2. Participation Grades

As can be seen in Figure 1, the participation grades of the co-researchers increased during the study from ‘co-option’ in the orientation phase to ‘collective action’ in the ‘data collection’ phase and onwards. ‘Co-learning’ was already achieved during the Kick-off workshop (setting-up phase). Additionally, it was possible to achieve a higher grade of participation in the course of one activity. For instance, during workshop 1, the grade increased from ‘compliance’ during the impulse lecture to ‘co-learning’ when discussing suitable data collection methods. However, to enable co-researchers to participate actively in the next step, it appeared to be obligatory that the AR took the lead by providing scientific knowledge.

Although the AR was responsible for the impulse lectures (see Table 1), the content of the presentations was, to an extent, influenced by the co-researchers. These lectures were necessary as this was the first PHR project in this department. Based on co-researchers’ needs, the AR decided what to include in the impulse lectures. For example, in the second lecture, various qualitative data collection methods were presented together with their benefits and limitations. Based on their input, co-researchers, CF and AR explored the advantages and disadvantages within their setting and decided on a suitable data collection method.

#### 3.3.3. Benefits and Limitations

The productive and constructive atmosphere, the transparency of the research process, the connectedness between the different professions, as well as the reflective nature and being appreciated by other co-researchers, were perceived as positive (Figure 2). On the other hand, PHR was experienced as very time consuming; the constant communication was demanding; the process was sometimes unpredictable which provoked uncertainty; the ‘black-box’ nature was challenging. With respect to the subject of the status-quo and needs assessment, the co-researchers appreciated the nearness of the research to the reality of their practical work, and the new perspectives they gained through the process. The structured way the AR organized the process was characterized as valuable. The AR experienced the process as motivating and empowering, e.g., understanding the problem from the inside. Giving up control and sharing the responsibility in the various study parts meant that the process required patience and tolerance from all team members, especially in the data collection phase. Additionally, flexibility of the entire team was essential.

#### 3.3.4. Reporting

The project was concluded with ‘reporting’, as a last and ongoing phase. All findings regarding the status quo and needs of the MDT with respect to holistic family caregiver support were included in a final document, which was presented in the hospital by the co-researchers together with the CF and the AR. Based on the outcomes, a ‘caregiver orientation’ folder was designed for the geriatric department to better inform caregivers regarding the different roles of the team and the caregiver offers within the department. Several co-researchers already presented outcomes and lessons learnt from their PHR experiences in practice on various occasions (e.g., on conferences), whereas CF and AR presented the outcomes and methodological considerations in scientific conferences.

## 4. Discussion

Our PHR study resulted in two outcomes, namely a comprehensive understanding of the actual state of the caregiver support within the geriatric acute and rehabilitation care, and the needs for offering comprehensive caregiver support within the department. The geriatric department needs a comprehensive, needs-oriented, personal, and professional caregiver support concept, which includes activities on both individual and system level. Personalized caregiver counselling is required to prepare caregivers for their new role and guarantee the quality of care in the home environment. Practical, accessible, and timely information provision for family caregivers must be developed. Our professionals (co-researchers) proposed to assess caregiver’s needs before the first counselling with a still to be developed assessment tool. Network building, communication and information materials are needed to enable organizational change. The hospital management should invest in a comprehensive caregiver support that goes beyond the usual economically focused case management routine. Resources (e.g., time, staff), and infrastructure (e.g., a counselling room), are required. Investing in health system development on a system level, e.g., offering capacity building activities and investing in intra-, inter- as well as extra-organizational networks, may be necessary. On an organizational level, caregivers must be acknowledged as part of the support team of geriatric patients.

Our outcomes are in line with conclusions from a qualitative literature review on hospital discharge planning practices for frail older people stating that there is room for organizational improvement by including the family, improving communication between professional health care employees and the family, better interdisciplinary communication and ongoing support after discharge. Specific contextual needs of family caregivers should be assessed, identification of resources that family caregivers need, and active and early involvement in the discharge process [51]. Our study also fitted recommendations to achieve a whole system change with respect to comprehensive, person-centered assessment and support for family caregivers towards the end of life, both on what needs to be in place within an organization to provide this support, e.g., a protocol for assessing caregivers and responding to the assessment, as well as on what is required to implement and sustain this approach in practice, e.g., a process for training practitioners and available time/workload capacity [52]. In a recent qualitative study from a Norwegian oncology setting, it was shown that health care workers in regional cancer care were also in need of organizational improvement, including systematic involvement of family caregivers in care pathways, and education of professional care workers in caregiver support [53].

The multidisciplinary team at the Rhein-Maas-Clinicum intends to use the findings of their study in the next step: building the new job of ‘Geriatric Family Companion’ [27]. However, to embed the new professional in the support system inside as well as outside the hospital, input of experienced family caregivers is needed and cooperation of professional support providers working in the area.

### Methodological Considerations

A major strength of our PHR study lies in the uniqueness of the applied strategy, as a truly partnered project. Participatory research provides a specific form of understanding that emerges from the synthesis of expertise brought to the partnership and the necessary negotiation of powers within that partnership [54]. This guarantees that the outcomes from the study are relevant, acceptable, and feasible (see e.g., [55]). Apart from the background research that was done by the AR, our co-research team was enabled to complete the whole research project. All function groups of employees within the German hospital setting that were in contact with family caregivers, were represented and contributed to the outcomes. Throughout the entire project, co-researchers were given the power to influence and shape the study itself. Through the active participation of professionals with different backgrounds and professional roles, the research question was explored and answered from different perspectives and its objectives were developed from the real-life experiences [56].

Participation cannot be imposed on research, but should rather be seen as an emergent process, developing from the interaction between the problem, the environment, and the aims, capacities and skills of those involved [35]. Participation depends upon the establishment of an environment of trust within the group [57]. Unfortunately, we did not formally study our group dynamics, e.g., with an instrument [58], so we cannot be sure that everybody’s viewpoints and expertise were valued in the process.

One of the other limitations of the PHR approach is that is has a gradualist tendency (evolution, not revolution) potentially leading to tunnel vision, caused by selection mechanisms [28]: the researchers have no way of knowing if what they discovered is actually true. However, since PHR is practice research, misinterpretations or gaps can still be uncovered and mended afterwards, in the phase of the programme’s implementation [28]. Nevertheless, based on the success of our project and the positive reflections by the co-research team, we feel confident that the outcome of the study reflects the actual status-quo and needs of a family caregiver support initiative.

The co-researchers felt empowered to continue with their newly acquired skills and knowledge. Our PHR project was terminated in January 2018, but some effects are still noticeable since the final meeting: co-researchers continue to participate in dissemination activities, e.g., (co-)authoring articles or presenting outcomes on conferences. In addition, a professional caregiver-working group was founded, and a ‘caregiver orientation’ folder for the geriatric department was designed.

Our PHR study was the first one applied in this hospital setting. As the German health system has been functioning historically through hierarchical decision-making [59], some challenges were expected. Interestingly, looking back on the PHR approach, our co-researchers and participants in the interviews and focus groups described PHR to be “*transparent, expedient, interconnecting, ‘eye-opening’, appreciative, and perspective-providing*” (final workshop, 1/2018). However, the PHR process was resource demanding. It took a vast amount of time (as can be seen in Table 1) and required extensive communication on different levels. In addition, due to the high workloads and understaffing within the department, the composition of the co-researchers in the research team varied over the project life cycle. The different professions in the MDT were represented in each meeting, but not always by the same persons. This situation led to time constraints, communication gaps and “perceived chaos” in the ‘Setting-up’, ‘Planning’ and ‘Data collection’ phases. With respect to the higher management level, a lack of interest and willpower to be involved was perceived. Despite several attempts, co-researchers were unable to schedule a meeting to present their outcomes to the management team. Even with the frustration caused by this, the higher hospital management was kept informed about the progress and invited to the meetings. PHR seemed to be an effective strategy within this hierarchically organized hospital. However, conducting PHR requires methodological experience, elaborated communication skills, a flexible time investment, and an open-minded atmosphere within the setting.

## 5. Conclusions

Through a participatory research methodology, a comprehensive understanding of the actual state of family caregiver support within the geriatric acute and rehabilitation care was gained. The geriatric department requires a comprehensive, needs-oriented, personal, and professional caregiver support concept, which includes activities on both individual and system level. To prepare family caregivers for their new role and guarantee the quality of care in the home environment, personalized caregiver counselling is necessary. Practical, accessible, and timely information provision for family caregivers must be established. To enable organizational change, network building, communication and information materials are needed, but above all, caregivers must be acknowledged as part of the support team of geriatric patients. On a system level, investing in health system development may be necessary.

Participatory health research appears to be a feasible and effective strategy within a hierarchically organized hospital. However, conducting a participatory methodology requires methodological experience, elaborated communication skills, a flexible time investment, and an open-minded atmosphere within the setting.

## Figures and Tables

**Figure 1 ijerph-17-05901-f001:**
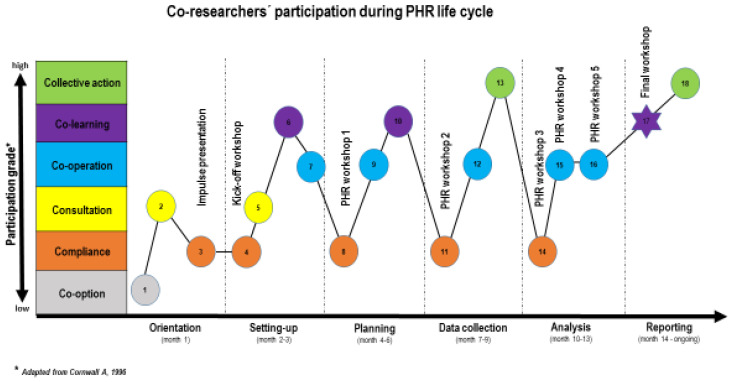
Co-researchers’ participation during PHR project, presented per phase in the project and per level of participation [43]. **Legend:** 1—literature review; 2—field visit, 3—impulse presentation; 4—impulse lecture PHR; 5—exploring expectations and goals; 6—identifying the problem and formulating research question; 7—forming the research team; 8—impulse lecture methodology, 9—exploring methodological advantages and disadvantages, 10—determining suitable data collection methods; 11—impulse lecture data collection; 12—determining plan of action and question catalogue; 13—collecting data; 14—impulse session data analyses and decision on ‘analyzing team’; 15—interactive data analyses; 16—interactive data analyses; 17—critical reflection of experiences with PHR; 18—dissemination activities.

**Figure 2 ijerph-17-05901-f002:**
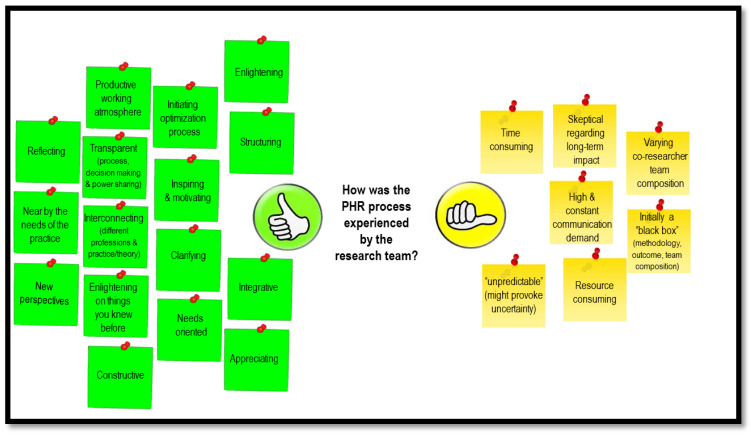
Perceived benefits and limitations of the PHR approach within the geriatric hospital setting: outcomes of the brainstorm session in the final meeting.

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
