# Peer review of "Caring for Family Caregivers of Geriatric Patients: Results of a Participatory Health Research Project on Actual State and Needs of Hospital-Based Care Professionals"

_ijerph, 2020, doi:10.3390/ijerph17165901_

Round 1

Reviewer 1 Report

I believe that research contributes to increasing knowledge about the study phenomenon. I would like to congratulate the authors for all the work done and, above all, for the practicality of the research. However, I consider that, in order to be published, the authors must improve several fundamental aspects.

Next, I will break down each of these aspects:

In line 32, the correlative numbers corresponding to the bibliographic references indicate them as [1,2], in line 41 as [8,9] separated by a specific spelling sign “,”. However, on line 46 they indicate it as [14-15], on line 48 as [6-7] and on line 52 as [19-20], separated by the sign “-“. It is necessary to review the entire document unifying the way of citing correlative references.

In line 34, I recommend deleting the phrase “(adapted from”, writing only the numbers [3, 4] that correspond to the bibliographic references.

In line 82, in section “2. Materials and Methods”, the authors do not indicate which methodological quality criteria in qualitative studies have guided their study, such as the COREQ criteria or the SRQR recommendations. I recommend that the authors review both documents and complement this section with considerations indicated in them. Also, if the magazine allows it, they can contribute additional material with tables indicating compliance or non-compliance with the different quality criteria.

https://www.equator-network.org/reporting-guidelines/coreq/

https://www.equator-network.org/reporting-guidelines/srqr/

In addition, they must indicate how they have ensured the rigor and methodological quality that govern qualitative studies: credibility, transferability, dependability and confirmability.

In line 113, they must correct the following sentence “(eg [34-35]”, since they do not close the parentheses.

In line 127, with respect to ethical considerations, they do not indicate that the study has been approved by the ethics committee of the hospital or another entity. Furthermore, they indicate that the consent of the study participants was obtained verbally. I consider that both aspects could imply the breach of the ethical principles of research with people. Authors should clarify these aspects.

In line 383, section “4. Discussion”, should be improved by discussing its results with studies of international relevance and not only in the aspect referred to “Methodological considerations”.

In line 408, they should correct the following sentence “(see f.i. [42]”, since they do not close the parentheses.

A cordial greeting.

Author Response

ijerph-885362/03-08-2020

Reply to Reviewer 1

I  believe that research contributes to increasing knowledge about the study phenomenon. I would like to congratulate the authors for all the work done and, above all, for the practicality of the research. However, I consider that, in order to be published, the authors must improve several fundamental aspects.

Dear reviewer, thank you very much for your kind words about our study and your very attentive look at our manuscript.  I have indicated the alterations in the following text point-by-point for as far as possible. In the manuscript I have used the track-changes option and highlighted new references.

In line 32, the correlative numbers corresponding to the bibliographic references indicate them as [1,2], in line 41 as [8,9] separated by a specific spelling sign “,”. However, on line 46 they indicate it as [14-15], on line 48 as [6-7] and on line 52 as [19-20], separated by the sign “-“. It is necessary to review the entire document unifying the way of citing correlative references.

All numbers corresponding to all references were reviewed and adjusted where necessary.

In line 34, I recommend deleting the phrase “(adapted from”, writing only the numbers [3, 4] that correspond to the bibliographic references.

Deleted the ‘adapted from‘ phrase in line 34

In line 82, in section “2. Materials and Methods”, the authors do not indicate which methodological quality criteria in qualitative studies have guided their study, such as the COREQ criteria or the SRQR recommendations. I recommend that the authors review both documents and complement this section with considerations indicated in them. Also, if the magazine allows it, they can contribute additional material with tables indicating compliance or non-compliance with the different quality criteria.

https://www.equator-network.org/reporting-guidelines/coreq/

https://www.equator-network.org/reporting-guidelines/srqr/

We added the following sentence:

All research choices, activities and reporting in this manuscript were guided by the Standards for Reporting Qualitative Research [46].

With reference to 46. O'Brien BC, Harris IB, Beckman TJ, Reed DA, Cook DA. Standards for reporting qualitative research: a synthesis of recommendations. Acad Med. 2014;89(9):1245-1251.

In addition, they must indicate how they have ensured the rigor and methodological quality that govern qualitative studies: credibility, transferability, dependability and confirmability.

Inserted the following text: Regular meetings between the AR and the supervisor (last author) were organized to discuss important research issues and the progress of the subsequent steps, but also ways of ensuring rigor of the study [44, 45].

With reference to: 44. Guba, E., & Lincoln, Y. Fourth generation evaluation. Newbury Park, CA, USA: Sage, 1989; and 45. Morse, J.M.  Critical Analysis of Strategies for Determining Rigor in Qualitative Inquiry. Qual Health Res 2015, 25, 1212–1222. https://doi.org/10.1177/1049732315588501.

In line 113, they must correct the following sentence “(eg [34-35]”, since they do not close the parentheses.

Added parenthesis

In line 127, with respect to ethical considerations, they do not indicate that the study has been approved by the ethics committee of the hospital or another entity. Furthermore, they indicate that the consent of the study participants was obtained verbally. I consider that both aspects could imply the breach of the ethical principles of research with people. Authors should clarify these aspects.

The text under 2.4. Ethical considerations was modified, explanations were added.

This participatory project aimed to assess the status-quo and needs of hospital personnel regarding support of family caregivers of geriatric patients, in preparation of their new to develop working practice. Participation as co-researcher or participant in one of the in-depth study parts was on voluntary basis, no incentives were offered. Verbal informed consent was obtained from all individual study participants in the professional peer groups before data collection. The head of the department enabled participation during working hours and facilitated the research activities (e.g. providing materials and interview rooms). Ethical approvement was given by the hospital management, approvement from an ethics committee was not necessary.

In line 383, section “4. Discussion”, should be improved by discussing its results with studies of international relevance and not only in the aspect referred to “Methodological considerations”.

New text was inserted, comparing our results with results from other studies:

Our outcomes are in line with conclusions from a qualitative literature review on hospital discharge planning practices for frail older people stating that there is room for organizational improvement by including the family, improving communication between professional health care employees and the family, better interdisciplinary communication and ongoing support after discharge. Specific contextual needs of family caregivers should be assessed, identification of resources that family caregivers need, and active and early involvement in the discharge process [51]. Our study also fitted recommendations to achieve a whole system change with respect to comprehensive, person-centered assessment and support for family caregivers towards the end of life, both on what needs to be in place within an organization to provide this support, f.i. a protocol for assessing caregivers and responding to the assessment, as well as on what is required to implement and sustain this approach in practice, f.i. a process for training practitioners and available time/workload capacity [52]. In a recent qualitative study from a Norwegian oncology setting it was shown that health care workers in regional cancer care were also in need for organizational improvement, including systematic involvement of family caregivers in care pathways, and education of professional care workers in caregiver support [53].

For this we added 3 extra references, two of them were reviews:

  1. Bauer, M.; Fitzgerald, L.; Haesler, E.; Manfrin, M. Hospital discharge planning for frail older people and their family. Are we delivering best practice? A review of the evidence. J Clin Nurs 2009, 18, 2539-2546. https://doi.org/10.1111/j.1365-2702.2008.02685.x.
  2. Ewing, G.; Grande, G.E. Providing comprehensive, person-centred assessment and support for family carers towards the end of life: 10 recommendations for achieving organisational change. London: Hospice UK, 2018.
  3. Røen, I., Stifoss-Hanssen, H., Grande, G. et al.Supporting carers: health care professionals in need of system improvements and education - a qualitative study. BMC Palliat Care 2019, 1858. https://doi.org/10.1186/s12904-019-0444-3.

In line 408, they should correct the following sentence “(see f.i. [42]”, since they do not close the parentheses.

Added the closing parenthesis

A cordial greeting.

On behalf of the research team: kind regards!

Reviewer 2 Report

Dear authors,

Thank you for the opportunity to review the manuscript entitled "Caring for family caregivers of geriatric patients: results of a participatory health research project on actual state and needs of hospital-based care professionals". This is a relevant work to implementing knowledge in a geriatric caregiving context through participatory health research method.

Although interesting, some revisions are needed to strengthen the manuscript.

  • On page 1, line 36  (about your citation): "Recent figures show that in Germany nearly 70% of hospitalized patients are depending on 36 support from their families when returning in their home environment [5]" -> This data is not recent, please modify the sentence
  • For the introduction section: on page 2, line 58. Specifically regarding "1.2. Context of the study":

    Please provide a citation to the reference methodologist at this stage (Spinuzzi);

    Please adequately report the reference inherent in Spinuzzi's article. 546 24. Spinuzzi, C. The Methodology of Participatory Design. Tech Commun 2005, 52, 162-174. -> This article is from 2005, and not from 2015

  • For the introduction section. Please improve the justify of the importance of the research in this context, before describing the study aim

  • For the aim of the study: on page 2, line 79. Please explicit the aim of the study distinctly. Enter the research question. Declare in distinct phrases what concerns the study from what interests the described process
  • For the Research strategy section. On page 2, line 84-97: Please simplify this step. Make it more understandable and smooth.
  • For Roles and activities. On page 3, line 98-113: Please make this step it more understandable and smooth. For example, better explains the professional figure of the Critical Friend
  • For the Research plan step. On page 3, line 115: Simplify this step. Remove the "(14 months)" specification.
  • For point 2. Materials and Methods: Please provide to insert an enlistment (for co-authors) and sampling section (for participants) here:

    Insert a co-authors description. Moreover, describe enlistment procedures, included if there was an explanation of the study, the project, etc.

    For participants: Please define the inclusion criteria or participants' characteristics you needed to involve specific key-informants.

  • For point 2. Materials and Methods: Please provide to insert a data collection section here:

    Describe the data collection strategy, and the reasons sustaining the data collection strategy. Discuss how the recruitment went; justify the setting for data collection. Explain how you conducted the data collection. Please explicit the form of data [for example, tape recordings, visual materials, field notes, observational notes, interviews' length, etc.].

  • For point 2. Materials and Methods: Please provide to insert a data analysis section here:

    Please describe the data analysis process. Provide to explain this process, and justify the choice

  • For 2.4. Ethical considerations section. On page 3, line 127: please improve this step. Ethical standards are not sufficiently explained. Please report if the ethics committee approved the study; or if the ethics committee did not require official approval for the study.
  • Please assess the utility of figure 1 (Figure 1. Picture taken at one of the meetings showing the PHR working practice.) for the purpose of scientific communication.
  • For the 4.1. Methodological considerations section. On page 17, line 403: Please, review this section also considering the reviews received.

Author Response

ijerph-885362/03-08-2020

Reply to Reviewer 2

Dear authors,

Thank you for the opportunity to review the manuscript entitled "Caring for family caregivers of geriatric patients: results of a participatory health research project on actual state and needs of hospital-based care professionals". This is a relevant work to implementing knowledge in a geriatric caregiving context through participatory health research method.

Dear reviewer, thank you very much for the appreciation of our manuscript and the recognition of the relevance of our study. Your comments were very helpful, the answers are outlined in the following text point-by-point.

Although interesting, some revisions are needed to strengthen the manuscript.

  • On page 1, line 36  (about your citation): "Recent figures show that in Germany nearly 70% of hospitalized patients are depending on 36 support from their families when returning in their home environment [5]"-> This data is not recent, please modify the sentence

The sentence was modified: Recent figures from the he German federal statistical office (Destatis) show that in Germany 76% of all care to those with care needs is provided by family caregivers in their home environment [5]. 

The reference [5] was changed to: Destatis: Pfegestatistik 2017. Pfege im Rahmen der Pfegeversicherung. Deutschlandergebnisse. Wiesbaden: Statistisches Bundesamt (2018)

  • For the introductionsection: on page 2, line 58. Specifically regarding "1.2. Context of the study":

Please provide a citation to the reference methodologist at this stage (Spinuzzi);

Inserted the reference to Spinuzzi in the Introduction/context part

Please adequately report the reference inherent in Spinuzzi's article. 546 24. Spinuzzi, C. The Methodology of Participatory Design. Tech Commun 2005, 52, 162-174. -> This article is from 2005, and not from 2015

Corrected the reference

  • For the introduction  Please improve the justify of the importance of the research in this context, before describing the study aim

The importance of our study was added as separate alinea in the Introduction:

Staff members at the Rhine-Maas-Clinicum treat family caregiver issues in an unstructured –haphazard- manner. The needs of family members, as prospective caregivers at home, are insufficiently acknowledged, although care professionals recognize the value of family members in the care trajectory of their patients. Family caregivers have been shown to play an unnamed but critical role in the geriatric health care system, forming a so-called ‘shadow workforce’ [23]. They are an important source of information for geriatric care professionals, being the ones able to supply key pieces of medical information about their elderly care receiver in a timely manner [23]. However, they often experience strain, especially co-resident caregivers, and run the risk of getting depressed [24]. In a large meta-analysis, it was shown that receiving support, formal or informal, was related to better caregiver health [25]. Supporting them will secure the care to geriatric patients in the home environment after hospital discharge [26].

With 4 extra references:

  1. Bookman, A.; Harrington, M. Family Caregivers: A Shadow Workforce in the Geriatric Health Care System? JHealthPolit Policy Law 2007, 32, 684-697. https://doi.org/10.1215/03616878-2007-040.
  2. Berg-Weger,M.; McGartland Rubio, D.; Tebb, S.S. Living with and Caring for Older Family Members. J Gerontol Soc Work 2000, 33, 47-62. https://doi.org/10.1300/J083v33n02_04.
  3. Pinquart,M.; Sӧrensen, S. Correlates of Physical Health of Informal Caregivers: A Meta-Analysis. J Gerontol B Psychol Sci Soc Sci 2007, 62, 126–137. https://doi.org/10.1093/geronb/62.2.P126.
  4. Bom, J.; Bakx, P.; Schut, F.; Van Doorslaer, E. The Impact of Informal Caregiving for Older Adults on the Health of Various Types of Caregivers: A Systematic Review. Gerontologist 2019, 59, e629–e642. https://doi.org/10.1093/geront/gny137.

  • For the aim of the study: on page 2, line 79. Please explicit the aim of the study distinctly. Enter the research question. Declare in distinct phrases what concerns the study from what interests the described process

The text was changed to:

1.3. Aim of the study

Before full implementation of the new family caregiver support concept in the Rhine-Maas-Clinicum, a pre-implementation step is necessary to gain insight in the status-quo activities and needs of the multidisciplinary team (MDT) regarding family caregiver support. Because of the haphazard way of engaging with family caregivers of geriatric patients by individual care professionals, little is known about how and what is needed to best deliver that support in a more organized way and to inform the hospital management on the best way of  implementing a new holistic support concept.

In this article, we report on both the process and the results of a PHR study on actual state and needs of the MDT. At the core of the PHR approach is the delegation of decision-making power regarding the research process from academic researchers to a co-research team consisting of members from the MDT. This co-research team will be formed at the start of the study and is responsible for choosing the research design, deciding on participant profile, data collection methods, data analysis and reporting. The actual research question will be articulated by this co-research team during the PHR process.

  • For the Research strategy On page 2, line 84-97: Please simplify this step. Make it more understandable and smooth.

The text was changed as follows:

2.1. Research strategy

For this pre-implementation step, a participatory research strategy is a preferred choice. PHR has a longstanding tradition in public health research [29], and is viewed with increasing interest by researchers, decision-makers, and practitioners worldwide as a strategy to overcome the gap between academic researchers and professionals working in practice [30]. One of the principles of PHR is a power shift in the conduct of research from academic or ‘expert’ researchers to those stakeholders in the field that possess insightful knowledge and experience [31,32]. Besides the generation of new knowledge, this approach also aims to empower those involved in the research process [33, 34]. Co-researchers learn methods of scientific knowledge production and use the process to understand their own reality, whereas academic researchers understand the real needs of a practiced based phenomenon [35]. Engaging members of the MDT as co-creator of new knowledge creates ownership of the outcomes [36].

  • For Roles and activities. On page 3, line 98-113: Please make this step it more understandable and smooth. For example, better explains the professional figure of the Critical Friend

The text was rewritten as follows:

2.2. Formation of the research team, roles, research activities

Three categories of researchers were involved in this study: academic researchers (AR), a critical friend (CF) and co-researchers.  The AR (first author) was responsible for: (1) facilitating and stimulating the PHR research process as “external impulse provider”, (2) contributing with scientific, technical and managerial knowledge, (3) drafting the research plan and monitoring the progress, and (4) guaranteeing a constant communication flow. The AR started the study by contacting all members of the MDT and explaining the participatory strategy.

During the research process, a ‘critical friend’ (CF, second author) supported the AR. A CF is a trusted professional who is able to ask proactive questions and give collegial feedback [37]. The CF in our study was serving as ‘listening ear’,  stimulated reflection and continuous professional development of the AR [38,39].

In our study, members of the MDT were participating as co-researcher with researchers from outside the setting. Members of the MDT are the ones with experiential knowledge in the setting, have ideas about relevant research questions, the acceptability and appropriateness of the research methods and the interpretation of the findings. In principle, all members from all professional groups of the MDT working at the geriatric department of the Rhine-Maas-Clinicum, were eligible as member of the co-research team provided that they have contact with family caregivers. Research experience was not necessary. The head of the department, a medical doctor, served as gatekeeper. The co-research team together with the AR and CF set the research agenda, formulate the research question, decide on which qualitative research methods to be used to collect the data, formulate selection criteria and recruit key informants in their peer group, collect the data, perform the data analyses, report the outcomes, and carry out all other necessary research activities to answer their research question [40,41].

  • For the Research plan step. On page 3, line 115: Simplify this step. Remove the "(14 months)" specification.

The 14 months specification was removed.

  • For point  Materials and Methods: Please provide to insert an enlistment (for co-authors) and sampling section (for participants) here:

Insert a co-authors description.

A co-author description can be found at the end of the manuscript, as required by the journal.

Moreover, describe enlistment procedures, included if there was an explanation of the study, the project, etc. For participants: Please define the inclusion criteria or participants' characteristics you needed to involve specific key-informants.

An explanation was inserted in the above-mentioned section 2.2. Formation of the research team, roles, research activities .A sentence was added to direct to the Results: :The outcomes of this process are reported in the Results section Part 1: PHR process, and in Table 1.

For point 2. Materials and Methods: Please provide to insert a data collection section here:

Describe the data collection strategy, and the reasons sustaining the data collection strategy. Discuss how the recruitment went; justify the setting for data collection. Explain how you conducted the data collection. Please explicit the form of data [for example, tape recordings, visual materials, field notes, observational notes, interviews' length, etc.].

The study participants and data collection/analysis methods were decided on by the co-research team. We described this in the Process part of the Results under the headings: Planning/data collection/analysis

  • For point  Materials and Methods:Please provide to insert a data analysis section here:
  • Please describe the data analysis process. Provide to explain this process, and justify the choice

As explained: the methods of analysis were decided on by the co-research team. We described this in the Results section under the headings: Planning/data collection/analysis

  • For 4. Ethical considerations section. On page 3, line 127: please improve this step. Ethical standards are not sufficiently explained. Please report if the ethics committee approved the study; or if the ethics committee did not require official approval for the study.

The text was adjusted:

This participatory project aimed to assess the status-quo and needs of hospital personnel regarding support of family caregivers of geriatric patients, in preparation of their new to develop working practice. Participation as co-researcher or participant in one of the in-depth study parts was on voluntary basis, no incentives were offered. Verbal informed consent was obtained from all individual study participants in the professional peer groups before data collection. The head of the department enabled participation during working hours and facilitated the research activities (e.g. providing materials and interview rooms). Ethical approvement was given by the hospital management, approvement from an ethics committee was not necessary.

  • Please assess the utility of figure 1 (Figure 1. Picture taken at one of the meetings showing the PHR working practice.) for the purpose of scientific communication.

Utility is low: we omitted the picture

  • For the 1. Methodological considerationssection. On page 17, line 403: Please, review this section also considering the reviews received

In this section we primarily discuss the PHR approach as a strategy to shift decision-making-power with respect to all research activities to members of the professional geriatric care team in the hospital without them having any research experience. They were expected to formulate their own research question and make all methodological choices to answer that question, had to do all the groundwork themselves as best as could be done. In the light of their inexperience we felt it not fair to discuss specific methodological choices (f.i had they not better used a different data collection method), that were essentially also the responsibility of the other members of the team (AR, CF, supervisor). The whole study was experienced by all as a participatory enterprise, build upon the interaction between those with and those without research knowledge.
